# γδ T Cells Mediate a Requisite Portion of a Wound Healing Response Triggered by Cutaneous Poxvirus Infection

**DOI:** 10.3390/v16030425

**Published:** 2024-03-10

**Authors:** Irene E. Reider, Eugene Lin, Tracy E. Krouse, Nikhil J. Parekh, Amanda M. Nelson, Christopher C. Norbury

**Affiliations:** 1Department of Microbiology & Immunology, Pennsylvania State University College of Medicine, Hershey, PA 17033, USA; 2Department of Dermatology, Pennsylvania State University College of Medicine, Hershey, PA 17033, USA

**Keywords:** poxvirus, γδ T cell, cutaneous infection, wound healing, IL-10, IL-22

## Abstract

Infection at barrier sites, e.g., skin, activates local immune defenses that limit pathogen spread, while preserving tissue integrity. Phenotypically distinct γδ T cell populations reside in skin, where they shape immunity to cutaneous infection prior to onset of an adaptive immune response by conventional αβ CD4^+^ (T_CD4+_) and CD8^+^ (T_CD8+_) T cells. To examine the mechanisms used by γδ T cells to control cutaneous virus replication and tissue pathology, we examined γδ T cells after infection with vaccinia virus (VACV). Resident γδ T cells expanded and combined with recruited γδ T cells to control pathology after VACV infection. However, γδ T cells did not play a role in control of local virus replication or blockade of systemic virus spread. We identified a unique wound healing signature that has features common to, but also features that antagonize, the sterile cutaneous wound healing response. Tissue repair generally occurs after clearance of a pathogen, but viral wound healing started prior to the peak of virus replication in the skin. γδ T cells contributed to wound healing through induction of multiple cytokines/growth factors required for efficient wound closure. Therefore, γδ T cells modulate the wound healing response following cutaneous virus infection, maintaining skin barrier function to prevent secondary bacterial infection.

## 1. Introduction

Skin-resident γδ T cells are among the first dedicated effector immune cells to encounter a cutaneous virus infection and have the potential to alter the course of the infection, prior to the expansion and infiltration of conventional T_CD4+_ and T_CD8+_ αβ T cells. Therefore, we sought to establish the role of skin-resident and skin-recruited γδ T cells upon control of a local cutaneous virus infection, along with their impact on systemic spread of the virus and the local pathology caused by the infection. Vaccinia virus (VACV) is an orthopoxvirus that was the most successful vaccine ever used (protecting against smallpox caused by variola virus) and now forms the basis for numerous clinical vaccine vectors. The efficacy of vaccination with VACV depends on cutaneous infection and damage [1]. There are numerous experimental and clinical indications that a cutaneous infection is the natural route of infection with VACV [2,3,4]. VACV is also related to both the skin-tropic orf virus, a prominent poxviral veterinary pathogen, and molluscum contagiosum, a poxvirus which infects the skin of tens of millions around the world and which can cause severe infections in immunocompromised patients [5,6]. Thus, understanding the mechanisms and interactions between VACV and the host will provide insights about similar interactions between human and veterinary pathogens and may identify therapeutic targets that allow for the manipulation of the host response, or which ameliorate pathology following infection. After systemic infection, γδ T cells can exhibit cytolytic control over the replication and spread of VACV infection [7,8,9,10] or potentially facilitate a protective T_CD8+_ response [11], but our previous work has identified key differences in how the immune system responds to control viruses after systemic infection versus the more relevant peripheral cutaneous infection [12,13,14]. In particular, we have identified both cellular and molecular mechanisms that are induced following cutaneous VACV infection which act to control the extent of local pathology, measured by lesion size and tissue loss, but which do not have large effects upon local VACV replication [12,13,14]. Therefore, we sought to examine the role of both skin-resident and skin-recruited γδ T cells in control of both virus replication and tissue pathology after cutaneous VACV infection.

Adult murine skin contains γδ T cell subsets that reside in both the epidermis and dermis and are distinguished by the Vγ TCR chain expression. Dendritic epidermal T cells (DETCs) express the Vγ3 TCR chain (Garman and Raulet nomenclature [15]), whereas dermal γδ T cells express either Vγ2 or Vγ4 [16,17]. Similar populations are present in human skin. Tissue-resident cells in the skin reside among epithelial cells and are distinguished by expression of the epithelial cell adhesion molecule (Ep-CAM, CD326). In various skin pathologies, γδ T cell subsets each have distinct functions, including production of IFN-γ [18,19,20,21,22] or IL-17A [23,24,25,26], recruitment of myeloid cells [27,28], lysis of virus infected cells [19,29,30], and pro-wound healing functions [27,31,32,33,34,35,36,37,38,39,40,41,42,43,44,45]. Therefore, it is likely that each resident γδ T cell type, along with recruited γδ T cells, has a differential ability to be activated and, correspondingly, has a different function, following cutaneous VACV infection. In addition, γδ T cell phenotype and function are also influenced by the local microbiome [46], adding an additional wrinkle to any phenotype or function observed in uninfected or virus-infected skin.

In this study, we found that, although Vγ3 DETCs display a cytolytic phenotype in response to cutaneous VACV infection, neither resident nor recruited γδ T cells are involved in the local control of VACV replication or in the control of spread of VACV systemically. However, in the absence of γδ T cells, VACV infected mice displayed a marked increase in tissue pathology in comparison to that observed in a WT situation. Neither tissue-resident nor recruited γδ T cells appear to modulate any of the previously defined tissue protective functions (e.g., recruitment of myeloid cell populations or the production of reactive oxygen species or Type I interferons) following VACV infection [12,13,14]. However, while investigating the mechanisms by which γδ T cells impact local tissue pathology, we discovered induction of a unique wound healing signature early after VACV infection. This signature bears hallmarks of the classical wound healing response found after sterile wounding, but the presence of an ongoing virus infection also induced some marked departures from the sterile wound healing response. Notably, this wound healing signature is induced early after virus infection, prior to the peak of virus replication in the skin, in contrast to the accepted paradigm, in which wound healing begins only after virus clearance [47,48]. Our data indicate that γδ T cells mediate a requisite component of this local viral wound healing response by producing, or inducing the production of, IL-17A, IL-22, IL-10, keratinocyte growth factor (KGF), and fibroblast growth factor 9 (FGF9). Therefore, the actions of γδ T cells are likely crucial in the closure of wounded skin following cutaneous virus infection, the prevention of secondary bacterial infections within virus-induced lesions, and, therefore, in the maintenance of the crucial barrier function of the skin.

## 2. Materials and Methods

### 2.1. Mice

C57BL/6 (wild-type, WT) mice were purchased from Charles River Laboratories or Jackson Laboratories. Breeding pairs of B6.129P2-Tcrd^tm1Mom^/J (TCRδ^−/−^) [49], IL-10/GFP (Vert-X) reporter mice [50], and IL-22/tdTomato (Catch22) reporter mice [51] were purchased from Jackson Laboratories. These mice were on a WT background after a minimum of 12 backcrosses to C57BL/6 and bred in a specific-pathogen-free animal facility at the Penn State Hershey College of Medicine. All animals were housed and cared for according to guidelines from the National Institutes of Health and the American Association of Laboratory Animal Care (AALAC). The Penn State Hershey College of Medicine Institutional Animal Care and Use Committee (IACUC) approved all animal experiments and procedures (Protocol #PRAMS200846111, approved December 2008, reviewed yearly since then and fully re-reviewed triennially).

### 2.2. Viruses and Infections

Stocks of VACV strain WR were produced in 143B TK^−^ cells and purified from cell lysate following ultracentrifugation through a cushion of 45% sucrose. VACV-GFP was previously described [52]. For intradermal (i.d.) infections, mice aged 7–10 weeks were anesthetized with ketamine/xylazine and injected with 10^4^ PFU of VACV in <10 μL in each ear pinna.

To monitor pathogenesis in the ears, ear thickness was measured using a 0.0001 in. micrometer (Mitutoyo, Aurora, IL, USA). Lesion progression and subsequent tissue loss were measured daily using a metric ruler. To measure titers of virus in vivo, ears and ovaries (the target organ of VACV systemic spread [13]) were freeze–thawed three times, homogenized, and sonicated, then titers in cell lysates were assayed by plaque assay on 143B TK^−^ cells, as previously described [53]. Plaques were counted two days later.

### 2.3. Cell Isolation and Flow Cytometry

Pairs of ears from each mouse were split into dorsal and ventral halves, minced, and digested in a solution of 1 mg/mL collagenase type XI (Sigma-Aldrich, St. Louis, MO, USA) in media supplemented with 2% FBS and 5 mM CaCl_2_ for 1 h at 37 °C, 5% CO_2_. Collagenase was quenched with media containing 5% FBS and 5 mM EDTA. Digested tissue was passed through 40 μm nylon cell strainers to create a single cell suspension. For intracellular cytokine staining and CD107a degranulation assays, cells from 3 pairs of ears were pooled and 10^6^ of those cells were stimulated prior to staining for flow cytometry (see below).

A blockade of FcR-mediated binding of mAbs, and the subsequent staining of cells, was performed in supernatant from flasks of 2.4G2 hybridoma cells supplemented with 10% normal mouse serum. All the following mAbs used were purchased from BD Pharmingen: CD45 (30-F11), CD3ε (145-2C11), TCRδ (GL3), CD4 (RM4-5), NK1.1 (PK136), CD19 (1D3), CD90.2 (53-2.1), CD11b (M1/70), Ly6C (AL-21), Ly6G (1A8), CD107a (1D4B), Granzyme B (GB11), IL-17A (TC11-18H10), TNFa (MP6-XT22), and IFN-γ (XMG1.2). We also utilized the following from Biolegend: Vγ2 TCR (UC3-10A6), Vγ3 TCR (536), and CD64 (X54-5/7.1). CD8α (53-6.7) and CD11c (N418) were obtained from eBioscience. In addition, PE-Cy7 conjugated streptavidin (eBioscience) was used to label biotin-conjugated antibodies. To stain for granzyme B, cells were stained for surface markers, fixed in 2% paraformaldehyde (Electron Microscopy Sciences, Hatfield, PA, USA), then permeabilized and stained intracellularly for granzyme B in 2.4G2 supernatant containing 10% normal mouse serum and 0.5% saponin (Sigma, St. Louis, MO, USA). Sample data were acquired on either an LSR II or LSR Fortessa flow cytometer (both from BD Biosciences, San Jose, CA, USA) and were analyzed using FlowJo software (Tree Star, Ashland, OR, USA).

### 2.4. Intracellular Cytokine Staining Assay

Single cell suspensions of ears were stimulated for 5 h at 37 °C, 5% CO_2_ with 50 ng/mL phorbol myristate acetate (PMA; Sigma) and 1 μg/mL ionomycin (Sigma), or were unstimulated in the presence of 10 μg/mL brefeldin A (BFA; Sigma). For T cell stimulations, lymphocytes were isolated using centrifugation over Lymphocyte Separation Medium (Cambrex, East Rutherford, NJ, USA) and were then stimulated for 4 h with 1 μM VACV peptide prior to the addition of BFA. The VACV-derived peptides B8R, A8R, A3L, A23R, K3L, A47L, A42R, A19L, and 10G2 have been previously described [54]. Following PMA/ionomycin or peptide stimulation, cells were blocked in 2.4G2 supernatant containing 10% normal mouse serum and were then stained for CD45, CD3ε, TCR δ, Vγ2 TCR, Vγ3 TCR, and CD4. Cells were fixed in 2% paraformaldehyde then permeabilized and stained for intracellular IL-17A and IFN-γ in 2.4G2 supernatant supplemented with 10% normal mouse serum and 0.5% saponin. The net frequencies and numbers of cytokine-positive T_CD8+_ were calculated by subtracting the unstimulated background response.

### 2.5. CD107a Degranulation Assay

Cells from ears were stimulated for 5 h at 37 °C, 5% CO_2_ with 50 ng/mL PMA and 1 μg/mL ionomycin, or were unstimulated in the presence of 1.5 μg/mL monensin (Sigma) and PE-conjugated rat anti-mouse CD107a. Following stimulation, cells were blocked in 2.4G2 supernatant containing 10% normal mouse serum and then stained for CD45, CD3ε, TCR δ, Vγ2 TCR, Vγ3 TCR, and CD8α.

### 2.6. Immunofluorescence Microscopy

Ears were harvested and embedded in Tissue-Tek OCT (Sakura Finetek) then rapidly frozen by immersion in liquid nitrogen-cooled 2-methyl butane and were kept at −80 °C overnight. Cryostat sections (10–12 μm) were cut at −20 °C, mounted on glass slides, air-dried for 2–3 h, fixed for 10–15 min in 1% paraformaldehyde (pH 7.4), air-dried again for 30 min, and stained with antibodies to TCR γδ, Ly6C, Ly6G, or CD8 (clones as above). A positive signal was revealed by subsequent staining with fluorescently labeled secondary antibodies. Staining was visualized using an Olympus 1 × 81 deconvolution microscope and Slidebook 5.0 digital microscope.

### 2.7. Quantitative PCR

Tissues were harvested, digested as above, and total RNAs were extracted using the RNeasy Plus Mini Kit (Qiagen, Germantown, MD, USA) with DNase treatment, according to the manufacturer’s protocol. For qPCR using Taqman Gene Expression Assays (Applied Biosystems, Waltham, MA, USA) or Universal Probe Library (Roche, Indianapolis, IN, USA), cDNA was prepared using the Hi-Capacity cDNA Synthesis Kit (Applied Biosystems). For qPCR using RT^2^ PCR Profiler Arrays (Qiagen), cDNA was prepared using the RT^2^ First Strand Kit (Qiagen). qPCR was carried out on a StepOnePlus (Applied Biosystems) with either the FastStart Universal Probe Master Mix (Roche) or RT^2^ SYBR Green qPCR Master Mix (Qiagen). For the Taqman and FastStart Universal Probe assays, changes in gene expression are expressed as fold changes using the ΔΔ^Ct^ calculation method against naïve mice of the same genotype, with *gapdh* as the housekeeping gene. For the RT^2^ PCR Profiler Array data, changes in gene expression are displayed as mean fold changes between groups of mice, relative to a panel of “housekeeping” genes. SYBR Green primers were as follows: *fgf7* forward 5′-ATAGAAACAGGTCGTGACAAGG-3′ reverse 5′-CAGACAGCAGACACGGAAC-3′ *fgf9* forward 5′-GTAGAGTCCACTGTCCACAC-3′ reverse 5′-CAACGGTACTATCCAGGGAAC-3′. Taqman primer/probe sets (Thermo Fisher, Waltham, MA, USA) were as follows: *il17a* (Mm00439618_m1), *il22* (Mm00444241_m1), *gapdh* (Mm99999915_gl), and *ifng* (Mm01168134_m1).

## 3. Results

### 3.1. Infected and Uninfected γδ T Cells in the Skin Are Present in the Foci of VACV Infection and Expand Early after Infection

VACV is a widely used vaccine vector and cutaneous administration of the vaccine most effectively induces a protective adaptive immune response [1]. However, major complications of immunization with VACV arise either from uncontrolled virus replication or from uncontrolled inflammation at the site of infection. The immune cells that are present in the skin at the site of infection likely play a vital role in local control of virus replication or inflammation, or both, prior to the recruitment of innate and adaptive effector cells. γδ T cells reside in the dermis and epidermis of the skin [55], and mice lacking γδ T cells are deficient in control of systemic VACV infection [7,8,9,10,11] and other poxviruses [56,57,58,59,60]. To visualize the interaction of γδ T cells and VACV within the skin, we inoculated WT mice with VACV-GFP using a bifurcated needle, a method that both generates easily identifiable foci of infection and mimics the route of human immunization with VACV. We harvested ear tissue 4d after infection and stained tissue sections for the presence of TCRδ using antibodies. Using immunofluorescence microscopy, we observed significant numbers of TCRδ^+^ γδ T cells that display typical dendritic morphology within the GFP^+^ VACV lesion, as well as some cells localized along the boundaries of the lesions (Figure 1A). No staining for TCRδ was seen in control TCRδ^−/−^ mice (Figure 1B). The GFP fluorescence in Figure 1B marks the extent of the lesion and does not represent an increase in virus infection (see below). Thus, γδ T cells are in a position to sense VACV infection, to potentially lyse VACV-infected cells, and to orchestrate the subsequent innate and adaptive immune response. VACV appeared to infect some γδ T cells within the infected foci. To confirm that VACV did infect γδ T cells in the skin, we infected mice intradermally with VACV, harvested ears 4d post-infection, and analyzed γδ T cells using flow cytometry. This route of infection results in a localized infection in immunocompetent mice [3,4]. We observed that a proportion (~11–15%) of γδ T cells in the ear expressed GFP after infection with VACV-GFP, indicating that these cells become infected with VACV (Figure 1C).

To dissect γδ T cell responses in the skin following VACV infection, we stained single cell suspensions harvested from infected ears with antibodies to TCRδ, as well as with antibodies to conventional αβ T cells, and performed flow cytometry analysis using the gating strategy outlined in Figure 1D. The numbers of γδ T cells increased early following infection (>2-fold) and peaked 2d post-infection (p.i.) (Figure 1E). The number of γδ T cells did not increase after the injection of diluent alone. The increase in number of γδ T cells appeared to represent an expansion or recruitment of all the subsets of these cells. Notably, γδ T cells were the major lymphoid population in the VACV-infected ear until 4–5d p.i. (Figure 1E), as NK cells are not recruited to the site of infection [12]. In addition, conventional T_CD4+_ and T_CD8+_ were recruited to the ear from the naïve precursor pool but did not surpass the number of γδ T cells until at least 5d post-infection (Figure 1E). Indeed, although numerous studies have examined the T_CD8+_ response to VACV following infection of the skin [61,62], these cells were in the minority compared to T_CD4+_ cells, which accumulate with similar kinetics and persist for much longer following infection (Figure 1E).

To further define the γδ T cell populations in infected and uninfected skin, we quantified the number of Vγ3^+^ DETCs [15] and Vγ2^+^ dermal γδ T cells [16,17] or Vγ3^−^Vγ2^−^ γδ T cells. After VACV infection, staining with antibodies to Vγ3 and Vγ2 TCR chains revealed that DETCs, which were consistently more abundant than other γδ T cell subsets, nearly doubled in number 2d p.i. and then underwent a drastic contraction, before recovering following resolution of infection (Figure 1F). In contrast, Vγ2^+^ dermal γδ T cell numbers peaked somewhat later, around 4d p.i., but underwent a slow, gradual contraction (Figure 1G). On the other hand, the more abundant Vγ2^−^Vγ3^−^ dermal γδ T cell population, which likely expresses the Vγ4 chain, for which there is no commercially available antibody [17], displayed similar patterns of expansion, contraction, and recovery as DETCs (Figure 1H). The distinct expansion and/or recruitment kinetics of γδ T cell populations suggests that each population may have a distinct function and role that contributes to the overall successful host response to viral infection.

### 3.2. A Subset of γδ T Cells Displayed Cytolytic Function after VACV Infection but Did Not Control the Local Virus Replication or Systemic Spread

γδ T cells exhibit a variety of functions depending on the subtype, the stage of life at which they develop, and their sub-anatomical location [63]. As γδ T cells have previously been proposed to mediate antiviral immunity via cytolytic activity [19,29,30], we initially examined the ability of γδ T cell subsets (Vγ3^+^ DETCs, Vγ2^+^ dermal γδ T cells, or Vγ3^−^Vγ2^−^ γδ T cells), conventional T_CD8+,_ or T_CD4+_ from the site of cutaneous infection with VACV to degranulate in response to a broad activation signal, PMA/ionomycin treatment (Figure 2A). Degranulation, measured via cell-surface expression of the endosomal marker CD107a in response to PMA/ionomycin, was measured 4d and 8d post-infection, except in T_CD8+_, which were only present in the VACV-infected ear 8d post-infection. We found minor cell surface staining with an anti-CD107a antibody after PMA/ionomycin treatment in T_CD4+_ and Vγ3^+^ DETCs 8d post-infection, but a marked activation-induced degranulation by the majority of Vγ3^+^ DETCs four days after VACV infection (Figure 2A). We saw only background levels of CD107a staining in the other cell types examined, either with or without of PMA/ionomycin activation. Effective cytolysis of virus-infected cells often requires the secretion of serine proteases, such as granzyme B (GzB), during degranulation [64]. Therefore, we also examined intracellular expression of GzB by Vγ3^+^ DETCs, Vγ2^+^ dermal γδ T cells, or Vγ3^−^Vγ2^−^ γδ T cells, as well as by conventional T_CD8+_ or T_CD4+_, as above (Figure 2B). Eight days post-infection, approximately one-third of Vγ3^+^ DETCs, about one-half of T_CD4+_, and almost all of the T_CD8+_ expressed GzB, indicating they have the ability to display cytolytic activity against VACV-infected cells (Figure 2B). Earlier after infection, on day four, a similar proportion of about one-third of Vγ3^+^ DETCs expressed GzB, and a small number of T_CD4+_ also had cytolytic capability (Figure 2B). However, the majority of γδ T cells did not express GzB (Figure 2B). Taken together, these data indicate that, of the γδ T cells present, only Vγ3^+^ DETCs likely have the capability to directly kill VACV-infected cells in the ear, although these data stop short of supporting or disproving a role for the direct Vγ3^+^ DETC-mediated cytolysis of VACV-infected cells in vivo.

The observation that DETCs could possess cytolytic activity after VACV infection drove us to examine whether γδ T cells play a direct antiviral role following cutaneous infection, as previously proposed following systemic infection [7,8,9,10]. We examined VACV titers in the ear of WT and TCRδ^−/−^ mice, which lack γδ T cells, 5d post-infection (Figure 2C), a time point corresponding to the peak of virus replication [12] and 8d post-infection (Figure 2D), a time point before tissue loss occurs. VACV titers in the ear in TCRδ^−/−^ versus WT mice displayed only small (~1.5-fold) differences (Figure 2C,D), indicating that it is unlikely that γδ T cells significantly contribute to the control of local cutaneous VACV replication, either directly or indirectly.

A major function of the initial immune response to a peripheral virus infection is to contain the infection at the initial site, prior to recruitment of innate and adaptive effector cells, which then mediate clearance of the infection. Dermal VACV infection remains primarily localized to the ear following infection [3], but removal of various immune components can allow systemic spread to the primary site of VACV replication, the ovaries [13]. However, we found no significant spread of VACV to the ovaries of TCRδ^−/−^ mice, indicating that γδ T cells are likely not playing a critical role in restricting systemic spread of VACV (Figure 2C,D). Therefore, γδ T cells are not required to control local VACV replication or systemic spread of VACV.

Mice infected in the ears with VACV develop visible lesions that undergo necrosis and the necrotic tissue is then lost [3,4]. We have previously described a role for two recruited populations of monocytes, production of reactive oxygen species, and local Type I IFN production in control of the severity of pathology observed following dermal VACV infection [12,13,14]. However, none of the factors mentioned above are involved in control of local VACV replication or spread from the original site of infection. Therefore, we sought to examine whether γδ T cells in the skin may play a similar role in controlling the severity of pathology following VACV infection, without displaying a role in control over local VACV replication. By measuring tissue swelling, the dimensions of the lesions and ensuing tissue loss in WT and TCRδ^−/−^ mice, we assessed the extent of tissue damage following VACV infection. TCRδ^−/−^ mice exhibited no change in swelling of VACV-infected ear tissue in the six days following the initial infection when compared to that observed in WT mice (Figure 2E). However, TCRδ^−/−^ mice subsequently displayed visibly larger skin lesions than their VACV-infected WT counterparts 8d post-infection (Figure 2F,G); this was quantified to indicate a statistically significant change in lesion size from day 6–14 post-infection (Figure 2H). Correspondingly, there was also a visible and quantifiable acceleration and increase in the severity of tissue loss in TCRδ^−/−^ mice compared to their WT counterparts (Figure 2I–K). Taken together, these data indicate that, although they do not significantly alter the replication of VACV in the ear, γδ T cells do control the severity and progression of tissue pathology at the cutaneous site of VACV infection.

### 3.3. γδ T Cell Responses to VACV Do Not Influence the Local or Systemic T_CD8+_ Response to VACV

The observation that γδ T cells do not control VACV replication or spread but do ameliorate exacerbated tissue loss following VACV infection suggested that an overly zealous local immune host response in the absence of γδ T cells could cause an increase in local necrosis, in the absence of an increase in VACV replication. Such an observation would indicate an immunoregulatory or immunosupressive role for γδ T cells. To test whether this is the case, we examined the effect of γδ T cells upon multiple components of the cutaneous immune response following VACV infection. T_CD8+_ are a vital component of antiviral immunity that are recruited to the VACV-infected ear (Figure 1E) and localize around the periphery of the VACV lesion, thus preventing spread of the virus 4d post-infection [13,61,65]. A recent publication has indicated that, following systemic VACV infection, γδ T cells may be an important component in the induction of protective T_CD8+_ [11]. Therefore, we sought to examine a potential role for γδ T cells in the priming, recruitment, localization, or subsequent function of T_CD8+_ in the anti-VACV response following cutaneous infection. We found no differences in the numbers of T_CD8+_ recruited to the ear either 5d (Figure 3A) or 8d (Figure 3B) after cutaneous VACV infection of WT or TCRδ^−/−^ mice. We then examined the localization of T_CD8+_ relative to VACV-infected cells after VACV-GFP infection of WT or TCRδ^−/−^ mice (Figure 3C,D). In both WT and TCRδ^−/−^ mice, we observed T_CD8+_ around the periphery of the VACV lesion (Figure 3C,D), indicating that there was no defect in T_CD8+_ localization in the absence of γδ T cells. Next, we examined the production of IFN-γ and TNFα by T_CD8+_ in the VACV-infected ears of both mouse strains 5d (Figure 3E) or 8d (Figure 3F) post-infection. We found, on both 5d and 8d post-infection, that cytokine production by T_CD8+_ directly ex vivo was indistinguishable in WT vs. TCRδ^−/−^ mice (Figure 3E,F). These data suggest there is no overarching defect in the function of T_CD8+_ in the absence of γδ T cells following cutaneous VACV infection.

Up to this point, we have examined the recruitment, localization, and function of bulk T_CD8+_. It was possible that the study of bulk T_CD8+_ of many specificities obscured subtle changes in some antigen-specific T_CD8+_ populations. However, it was not possible to examine the function of antigen-specific T_CD8+_ at the site of infection, as many cells had already been activated in the presence of VACV-infected cells (Figure 3E,F). Therefore, we examined the antigen-specific systemic T_CD8+_ response in the spleen 8d after dermal VACV infection of WT or TCRδ^−/−^ mice by incubating splenocytes in the presence of nine individual MHC Class I-binding peptides derived from VACV. We measured production of IFN-γ and TNFα, each of which can exert antiviral effects and which, when both produced by the same cells, indicate polyfunctionality that correlates with protective capacity against virus infection [66] (Figure 3G,H). The proportion of T_CD8+_ producing IFN-γ in response to each epitope was greater than the proportion producing TNFα, but, across three replicate experiments, there were no statistically significant differences in the responses observed in WT and TCRδ^−/−^ mice (Figure 3G,H). Taken together, these results indicate that γδ T cells are unlikely to play a significant role in the priming, recruitment, localization, or subsequent function of T_CD8+_ in the anti-VACV response. Therefore, neither T_CD8+_-mediated immunopathology nor a defect in effective T_CD8+_-mediated immunity is likely to contribute to the increased pathology we observe in VACV-infected TCRδ^−/−^ mice.

### 3.4. γδ T Cells Do Not Influence the Local Monocyte Response to VACV

It is possible that, rather than controlling immunopathology, γδ T cells coordinate the changes in the immune response that lead to resolution of inflammation and subsequent reduction in local tissue pathology. We have previously described a role for two recruited populations of monocytes in control of the severity of pathology observed following dermal VACV infection, but these populations of monocytes do not have a large effect upon local VACV replication [12,13,14]. Both classical CD11b^+^Ly6C^+^Ly6G^−^ monocytes and a population of CD11b^+^Ly6C^+^Ly6G^+^ myeloid cells are recruited to the VACV lesion following i.d. infection [12,13,53]. The CD11b^+^Ly6C^+^Ly6G^+^ myeloid cells are monocytic in nature and limit local tissue damage via production of reactive oxygen species [12]. In contrast, the classical CD11b^+^Ly6C^+^Ly6G^−^ monocytes are attracted to the site of infection by Type I IFN-stimulated production of CCL4 and also moderate tissue damage, possibly by becoming infected and soaking up excess virions [14]. Since γδ T cells are found in the VACV lesion (Figure 1) and are known to regulate myeloid cell activity after some insults [27,28], we examined whether sensing of VACV by γδ T cells is required to moderate recruitment of either monocyte population. We harvested ears from infected WT or TCRδ^−/−^ mice and carefully gated monocyte populations to exclude innate lymphoid cells (ILCs), lymphocytes, or resident or recruited dendritic cells (DC) (Figure 4A). Five days after infection, a time point at which both virus replication and CD11b^+^Ly6C^+^Ly6G^−^ classical monocyte infiltration peak, there was no difference in recruitment of either classical monocytes (Figure 4B,C) or CD11b^+^Ly6C^+^Ly6G^+^ myeloid cells (Figure 4B,E) in VACV-infected WT compared to infected TCRδ^−/−^ mice. Eight days after infection, when numbers of CD11b^+^Ly6C^+^Ly6G^+^ myeloid cells peak, we also observed no difference in recruitment of either classical monocytes (Figure 4D) or CD11b^+^Ly6C^+^Ly6G^+^ myeloid cells (Figure 4F) in VACV-infected WT compared to infected TCRδ^−/−^ mice. In contrast to T_CD8+_, both CD11b^+^Ly6C^+^Ly6G^−^ monocytes and CD11b^+^Ly6C^+^Ly6G^+^ myeloid cells enter the VACV lesion, where the CD11b^+^Ly6C^+^Ly6G^−^ monocytes become infected and CD11b^+^Ly6C^+^Ly6G^+^ myeloid cells produce ROS to moderate tissue damage [12,65]. To examine whether γδ T cells control the localization of each myeloid cell population, we infected WT or TCRδ^−/−^ mice with VACV-GFP, harvested at 7d post-infection, and then stained for either Ly6C or Ly6G (Figure 4G–J). We observed similar staining with Ly6C and Ly6G, both inside and outside of the GFP^+^ VACV lesion, in both WT and TCRδ^−/−^ mice (Figure 4G–J). Therefore, our data indicate that γδ T cells do not play a requisite role in recruitment or localization of three crucial components of local cellular antiviral immunity.

### 3.5. IFN-γ Mediates Pathology in VACV-Infected Skin

Interferons are potent antiviral cytokines that are widely produced upon virus infection, but which are also strongly linked to the pathology of many skin conditions, including psoriasis [67] and alopecia areata [68]. We have demonstrated above that IFN-γ is produced by T_CD8+_ in response to cutaneous VACV infection, but that this process is not affected by the absence of γδ T cells (Figure 3A–H). However, direct production of IFN-γ by γδ T cells plays a role in control of multiple viral infections [18,19,20,21,22], including poxviruses following systemic infection [8,57,58,69]. Therefore, we examined the level of mRNA encoding IFN-γ in the VACV-infected ears at various times after infection. We found a robust (~40-fold) and reproducible induction of *ifng* transcript by three days post-infection (Figure 5A), a time point at which we have previously described there was little to no immune cell infiltrate present [12]. The induction of *ifng* transcript continued to rise to ~1–2000-fold within seven days of infection, likely as a result of infiltration of activated T_CD8+_ and T_CD4+_ (Figure 5A).

To examine the functional consequence of production of IFN-γ following cutaneous VACV infection, we measured the swelling of the VACV-infected ear at early time points in WT vs. IFN-γR^−/−^ mice. We found that, as early as 3d post-infection, VACV-infected IFN-γR^−/−^ mice displayed increased swelling compared to WT mice (Figure 5B). This correlated with development of a VACV lesion one day earlier in VACV-infected IFN-γR^−/−^ vs. WT mice (Figure 5C–E) and enhanced tissue loss at later time points (Figure 5F–H). IFN-γ is a potent antiviral cytokine that drives production of many interferon-stimulated genes that act to control virus replication using a large variety of mechanisms. However, the increased local tissue pathology we observed was not a result of enhanced VACV replication in the absence of the antiviral action of IFN-γ, as WT and IFN-γR^−/−^ mice displayed similar levels of replicating VACV at the site of infection in the ear 5d post-infection (Figure 5I). In addition, the absence of IFN-γ signaling did not alter spread of VACV from the ear, with similar low levels disseminating to the spleen and ovaries by day five post-infection (Figure 5J,K). Therefore, the phenotype we observe following VACV infection of IFN-γR^−/−^ mice, namely a marked increase in tissue pathology compared to VACV-infected WT mice, without a corresponding increase in VACV replication or spread, closely mimics the phenotype we observed following VACV infection of TCRδ^−/−^ mice.

To tease out the role of γδ T cell-mediated production of IFN-γ, we examined the production of IFN-γ 4d post-infection by staining for the intracellular cytokine, with or without activation of ex vivo, isolated γδ T cell subsets by PMA/ionomycin. Somewhat surprisingly, ex vivo, isolated γδ T cells of all kinds failed to stain for IFN-γ, even when activated with PMA-ionomycin for a number of hours before staining (Figure 5L). As a control for our activation and staining, we examined IFN-γ production by T_CD4+_ and T_CD8+_ harvested from the lymph node of VACV-infected mice 4d post-infection, a time point prior to infiltration of these cell types to the site of infection (Figure 5M). Although neither T_CD4+_ nor T_CD8+_ detectably produced IFN-γ in the absence of external stimuli, there was substantial and reproducible upregulation of IFN-γ production by both T_CD4+_ and T_CD8+_ upon PMA/ionomycin stimulation (Figure 5M), indicating that our assay was working.

Our observation that we could detect both *ifng* transcript (Figure 5A) and an increase in swelling in IFN-γR^−/−^ mice (Figure 5B), but no detectable production of IFN-γ protein by cutaneous γδ T cells (Figure 5J) at similar time points, could potentially be explained by the relative insensitivity of the flow cytometry staining for IFN-γ protein. Therefore, we examined the levels of *ifng* transcript in the ears of WT vs. TCRδ^−/−^ mice 4d post-infection, the day on which we had previously conducted the flow cytometry analysis. As above, we found a marked induction of *ifng* transcript upon VACV infection of WT mice, and this was enhanced slightly (~2-fold) in the absence of γδ T cells (Figure 5N). Therefore, γδ T cells do not contribute to IFN-γ production upon cutaneous VACV infection, a marked departure from their role following systemic infection [8,57,69,70].

Because γδ T cells are not required for IFN-γ production, and leukocytes are not recruited to the infected ear until after we began to observe induction of *ifng* transcript, we sought to identify the cell type producing IFN-γ early after dermal VACV infection. We infected mice, harvested cells from the ear 4d post-infection, incubated them in the presence of Brefeldin A for 6h to prevent protein secretion, stained cells with cell-surface markers to identify cell types, and then stained intracellularly to detect IFN-γ. We divided cells in the ear into EpCAM^−^ CD45^+^ infiltrating immune cells, EpCAM^+^ CD45^+^ resident immune cell populations (which include resident γδ T cells), and EpCAM^+^ CD45^−^ keratinocytes (KC). As in all of our flow cytometry experiments, we used fluorescence minus one (FMO) controls in which cells were stained with all antibodies except IFN-γ, in order to distinguish our IFN-γ signal. We found that neither EpCAM^−^ CD45^+^ nor EpCAM^+^ CD45^+^ (including resident γδ T) cells stained for the presence of IFN-γ 4d post-infection (Figure 5O,P). However, somewhat surprisingly, a small proportion of EpCAM^+^ CD45^−^ KC, which comprise the majority of cells in the ear, did produce IFN-γ (Figure 5O,P). Although the proportion of KC producing IFN-γ only increased from <0.1% of cells to ~0.54 +/− 0.07 of cells, KC outnumber all resident and recruited cells in the skin >100:1, so this increase in the number of cells is biologically significant.

### 3.6. γδ T Cell Subsets Are Responsible for IL-17 Production in VACV-Infected Skin

After we had shown that γδ T cells did not produce IFN-γ after cutaneous VACV infection, our attention turned to examine the production of other cytokines that have been implicated in skin pathology. In particular, our attention turned to the γδ T cell-mediated production of IL-17A, for a number of reasons. First, there is a strict division between the production of IL-17A and IFN-γ by γδ T cells [71] that may be controlled by the initiation of innate vs. adaptive signaling within these cells [22]. Second, γδ T cells are the primary producers of IL-17 outside of gut tissue [23,24], including in cutaneous infection models. Vγ2^+^ dermal γδ T cells are strong producers of IL-17A following BCG infection [72] and Vγ3^+^ DETCs produced IL-17A after cutaneous *S. aureus* infection, [25,26]. Third, γδ T cell-mediated production of IL-17A can also play a role in cutaneous pathology, driving tissue damage in psoriasis [17,73,74] and dermatitis [75]. Fourth, γδ T cell-mediated production of IL-17A can also play an important role in cutaneous wound healing [25]. Therefore, as γδ T cell-derived IL-17A clearly plays important roles in context-dependent anti-pathogen, as well as tissue damaging and tissue protective, responses, we examined the levels of *il17a* mRNA transcript upon cutaneous VACV infection. There was a marked (~10–15-fold) increase in *il17a* transcript in WT mice upon VACV infection (Figure 6A). When we examined levels of IL-17A protein produced in γδ TCR^+^ or γδ TCR^−^ cell populations using flow cytometry, using approaches similar to those outlined above, we found that >80% of the cells producing IL-17A were γδ T cells (Figure 6B) early after infection. We further examined the ability of γδ T cell subsets (Vγ3^+^ DETCs, Vγ2^+^ dermal γδ T cells, or Vγ3^−^Vγ2^−^ γδ T cells), to produce IL-17A directly ex vivo, or whether they possessed the ability to produce IL-17A, following activation with PMA/ionomycin. A small proportion of both the Vγ2^+^ dermal γδ T cells and Vγ3^−^Vγ2^−^ γδ T cells produced low quantities of IL-17A directly ex vivo (Figure 6C,D), and this production was enhanced by activation with PMA-ionomycin (Figure 6C). However, the Vγ3^+^ DETCs, which were the cells responding to VACV infection by upregulating cytolytic activity, did not produce IL-17A (Figure 6C,D), further indicating functional specialization of γδ T cell subsets upon dermal VACV infection. To assess the contribution of Vγ2^+^ and Vγ3^−^Vγ2^−^ γδ T cells to the production of IL-17A following VACV infection, we measured transcript levels of *il17a* mRNA in uninfected or infected WT or TCRδ^−/−^ mice, 5d post-infection. The increase in *il17a* transcript in WT mice infected by VACV was almost completely ablated in VACV-infected TCRδ^−/−^ mice (Figure 6E), indicating that γδ T cells, directly or indirectly, are likely responsible for almost all IL-17A production upon virus infection.

### 3.7. Dermal VACV Infection Induces a Response Characteristic of Wound Healing

Our data up to this point indicated that both γδ T cells and IFN-γ can have a profound impact upon local tissue pathology following VACV infection, without a large effect upon local VACV replication or systemic spread of the virus. This phenotype is reminiscent of those we have observed in mice depleted of either of two different monocyte populations [12,53], which lack myeloid cell production of reactive oxygen species [12], or mice lacking Type I IFN signaling [14]. Therefore, we took a step back to examine the processes induced at the site of dermal infection with VACV. We assessed the expression of various cytokines, chemokines, interferons, interferon receptors, and their signaling pathways in the skin of uninfected WT mice compared to VACV-infected WT mice 5d post-infection, a time point immediately prior to the development of lesions and subsequent tissue loss. This analysis did not directly address the role of γδ T cells, as we only examined modulation of gene expression by VACV infection in WT mice, but it did provide us with significant information with which to interrogate the role of γδ T cells (see below). In total, our gene profiling examined the expression of 160 unique transcripts, 72 of which (45%) were statistically changed more than 2-fold (50 upregulated, 22 downregulated) upon VACV infection (Figure 7A). Our focused approach allowed us to identify that ~76–77% of the 50 upregulated and 22 downregulated transcripts had a defined role in the process of cutaneous wound healing (upregulated wound healing genes in red and downregulated wound healing genes in blue in Figure 7A) [76,77,78], producing a putative gene signature for a virus-induced wound healing response in skin, shortly after infection.

Of the transcripts that were modulated by VACV infection, and have a role in cutaneous wound healing, were CC chemokines (Figure 7B) and CXC chemokines, which function to recruit innate and adaptive immune effector cells, all of which were upregulated. There were also marked upregulations of IL-10 superfamily members, including *il10*, *il22*, and *il24* (which was upregulated ~400-fold), and downregulation of the antagonist of the IL-10 superfamily member IL-20, *il20ra* (Figure 7C). Transcripts for other cytokines with a positive role in wound healing, such as *il1b*, *il6*, and *il9*, were upregulated, while cytokines that have an inhibitory role in wound healing, such as *il16* and *il4*ra, were downregulated upon VACV infection (Figure 7D).

Growth factors such as vascular endothelial growth factor (*vegf*), Leukemia inhibitory factor (*lif*), and osteopontin (*spp1*) were also upregulated by VACV infection. Somewhat surprisingly, transcripts of receptors for a number of growth factors involved in wound healing, including the growth hormone (*ghr*), leptin (*lepr*), and erythropoeitin (*epor*) receptors, were downregulated in response to VACV infection, along with the hormones thrombopoietin (*thpo*) and adiponectin (*adipoq*) (Figure 7E). This indicates that the presence of VACV infection in the skin may modulate the classical wound healing response, creating a unique poxvirus/wound healing signature. In addition, members of the Transforming Growth Factor-β superfamily (*tgfb2*, *bmp2*, *bmp4*, and *bmp6*) that play a role in fibrosis and scar formation in the classical wound healing response [79,80] were also downregulated upon VACV infection (Figure 7F), further indicating a departure from the classical response.

### 3.8. γδ T Cells Modulate Expression of Genes Involved in Wound Healing after VACV Infection

We expanded on our findings to examine the impact of a constitutive deficiency in γδ T cells upon the expression of cytokines and chemokines identified above as being involved in the wound healing process initiated by dermal VACV infection. As above, we utilized qPCR arrays to profile gene expression 5d post-infection. Importantly, in order to rule out any contribution of constitutive changes in gene expression in TCRδ^−/−^ mice, we initially analyzed gene expression in these mice compared to WT mice, in the absence of infection. We found that a number of transcripts were upregulated in uninfected mice in the absence of γδ T cells, including *cxcl13*, *il23a*, and *il12b* (Figure 8A). We also found that *adipoq* and *il16*, which are genes involved in wound healing that we have previously shown to be modulated during VACV infection, were constitutively downregulated in the skin of TCRδ^−/−^ mice, implying that a constitutive alteration in wound healing may exist in these mice (Figure 8A). However, as we observed a change in *adipoq* and *il16* in the absence of infection in TCRδ^−/−^ mice, we could not draw any conclusions about the role of these genes in the response after VACV infection. We next examined changes in gene expression in uninfected vs. infected TCRδ^−/−^ mice to reveal VACV-induced changes in gene expression in these mice. We found a similar pattern of gene expression changes to that shown in Figure 7A, with both upregulation and downregulation of numerous cytokines and chemokines, some of which are involved in wound healing (upregulated wound healing genes in red and downregulated wound healing genes in blue, in Figure 8B). Therefore, to examine the contribution of γδ T cells to gene expression exclusively after VACV infection, we compared gene expression in WT vs. TCRδ^−/−^ mice 5 days after dermal VACV infection, a time point at which there is no significant difference in titers of VACV within the infected ear (Figure 2D). We found that a number of transcripts are induced to higher levels in TCRδ^−/−^ mice than WT mice, including a number of interferon-responsive chemokines such as *cxcl9*, *cxcl13*, and *ccl5* (Figure 8C). This correlates with enhanced expression of *ifna2* observed in VACV-infected TCRδ^−/−^ mice (Figure 8B). However, we also observed that two IL-10 superfamily members, *il10* and *il22*, which were upregulated upon dermal VACV infection (Figure 7A), were not upregulated in VACV-infected TCRδ^−/−^ mice (Figure 8B) and were markedly (8–15-fold) and significantly downregulated in VACV-infected TCRδ^−/−^ mice vs. VACV-infected WT mice (Figure 8C). Notably, neither the expression of *il10* nor *il22* was modulated in uninfected TCRδ^−/−^ mice (Figure 8A), indicating that the deficiency in *il10* and *il22* expression was both γδ T cell- and VACV-dependent.

To examine whether γδ T cells express IL-22 directly, or indirectly cause it’s production, we utilized Catch22 mice, in which the IL-22 promoter drives expression of the fluorescent reporter molecule, tdTomato [51]. We infected these mice i.d. with VACV expressing GFP, harvested cells 5d post-infection, and assessed tdTomato fluorescence in GFP^+^ and GFP^−^ cell populations liberated from the ear. As expected, and as previously published [65], we found that a large number (~7%) of EpCAM^+^CD45^−^ KCs were VACV-infected, but we found no tdTomato fluorescence in either infected or uninfected KC populations (Figure 8D). We next separated immune cell populations present at the site of VACV infection into resident (EpCAM^+^CD45^+^, Figure 8E) or recruited (EpCAM^−^CD45^+^, Figure 8G) populations. Both resident and recruited populations displayed significant levels of infection (GFP fluorescence, Figure 8E,G), but the resident population displayed a much higher proportion of cells displaying IL-22 production than the much more numerous recruited population (tdTomato fluorescence, Figure 8E,G). Within the resident immune cells, there were distinct populations of cells that were either infected (GFP^+^), infected and producing IL-22 (GFP^+^ tdTomato^+^), or uninfected but also producing IL-22 (GFP^−^ tdTomato^+^) (Figure 8E). To ascertain the contribution of resident γδ T cells to IL-22 production, we also stained cells with an anti-γδ TCR antibody and examined IL-22 production by resident (Figure 8F) and recruited (Figure 8I) γδ T cells or other CD45^+^ immune cells (Figure 8H). We were able to observe VACV infection of resident EpCAM^+^ γδ T cells (Figure 8F), as previously (Figure 1C), and a large population of IL-22-producing resident γδ T cells (Figure 8F), but production of IL-22 by VACV-infected γδ T cells was minimal (Figure 8F). Therefore, it is likely that other resident CD45^+^ cells that are VACV-infected also contribute to production of IL-22. Within the recruited (EpCAM^−^) immune cell populations, IL-22 production has often been attributed to a population of T_CD4+_, so we examined VACV infection and IL-22 production in γδ TCR^−^ lymphocyte populations (Figure 8H) and compared this to infection of, and IL-22 production by, recruited γδ T cells (Figure 8I). We found that infection of both populations of recruited lymphocytes was minimal (Figure 8H,I), but that recruited γδ T cells, but not other lymphocytes, also produced IL-22. Finally, we examined the contribution of γδ T cells to *il22* mRNA levels 5d post-VACV infection of the ear, and found that, although there was a slight increase in *il22* transcript levels in VACV-infected TCRδ^−/−^ mice compared to uninfected TCRδ^−/−^ mice, this induction was markedly and statistically significantly lower than the induction we observed in VACV-infected WT mice (Figure 8J). Taken together, the data from Figure 8D–J indicate that both resident and recruited γδ T cells are major producers of IL-22 during dermal VACV infection, but that the IL-22 producing γδ T cells are primarily not VACV infected.

To examine whether γδ T cells express IL-10, the other molecule we identified as being modulated in the ear by VACV infection, we utilized reporter mice in which an internal ribosome entry site (IRES)-enhanced green fluorescent protein (*eGFP*) fusion protein was placed downstream of exon 5 of the interleukin 10 (*Il10*) gene (Vert-X mice) [50]. We were able to detect a small but distinct and reproducible population of cells liberated from the VACV-infected ear that were GFP^+^ 5d post-infection of the IL-10-GFP reporter mice (Figure 8L), but not of WT mice (Figure 8K). CD45^−^EpCAM^+^ KCs did not contribute to the IL-10-GFP signal observed (not shown), and the day five time point examined is prior to accumulation of antigen-specific T_CD8+_ [12,53] that have previously been shown to produce IL-10 [81], so we examined IL-10 production by resident (EpCAM^+^) or recruited (EpCAM^−^) γδ T cells or myeloid cells, as previously identified in Figure 4. In contrast to the γδ T cell production of IL-22 observed above (Figure 8F,I), we did not find IL-10 production by either resident or recruited γδ T cell populations (Figure 8M,O). In contrast, we did find production of IL-10/GFP by resident, but not recruited, myeloid cells populations (Figure 8N,P). Therefore, γδ T cells do not appear to express IL-10 directly upon VACV infection but may modulate it’s expression in other ways.

Our identification of γδ T cell-modulated genes is a minimalistic one, so we sought to examine a number of other genes that have been linked to both γδ T cells and wound healing. Two of these genes, Fibroblast growth factor 9 (*fgf9*, Figure 8Q [40]) and Keratinocyte Growth Factor (*fgf7*, Figure 8R [42]) were upregulated in the ear >40-fold 5d post-VACV infection in WT mice. However, *fgf9* was not upregulated above the levels observed in uninfected mice in VACV-infected TCRδ^−/−^ mice (Figure 8Q) and the upregulation of *fgf7* was markedly reduced (Figure 8R). Therefore, γδ T cells directly or indirectly modulate the expression of multiple genes involved in the wound healing program initiated by cutaneous VACV infection.

## 4. Discussion

It is generally accepted that the local innate immune response to a peripheral virus infection is designed to slow virus replication and spread until an adaptive response can be initiated that will eliminate the virus and prevent re-infection. However, a key factor governing the extent of both the innate and adaptive immune response is that neither response should deleteriously affect the host, making it more susceptible to secondary infections, particularly via loss of barrier functions in the periphery (skin, airways, etc.). Typically, this is represented in textbooks as a temporally regulated process, with the immune system gaining control over the virus, followed by a subsequent switch to reparative mechanisms that ameliorate tissue damage inflicted by both the virus and the ensuing host response. However, in this report we have described the initiation of a wound healing program in the skin, concurrent with the deployment of innate antiviral strategies. The action of this wound healing program can alter tissue pathology following virus infection, independent of the control of virus replication and spread. The presence of two ongoing responses at the same time, and in very close physical proximity to each other, raises the possibility that the antiviral and wound healing responses could act synergistically to enhance each other by increasing the efficiency of monocyte recruitment or interferon production (see below). However, it is also possible that components of one response will act to decrease the efficiency of the other. In either case, it is clear that wound healing after a cutaneous virus infection differs mechanistically from sterile wound healing, creating novel points of therapeutic intervention that could enhance wound healing and successful closure of the barrier surface to prevent secondary bacterial infection.

In a number of previous manuscripts, we, and others, have described a role for various aspects of the local innate immune response, including two distinct monocyte populations [12,65], reactive oxygen species [12] and Type I IFN [14], in amelioration of tissue damage following peripheral VACV infection. However, in each of these publications we found little or no effect of these components of the innate immune response upon local virus replication, despite often profound effects upon local tissue pathology [12,14,65]. Having outlined a wound healing program initiated upon cutaneous virus infection here, we can now retrospectively place each of these innate immune responses into that program. Recruitment of monocytes to a wound is required for effective wound healing to occur [82,83], as is the Type I IFN receptor [84,85]. Similarly, Ly6G^+^ cells are recruited to a wound [86] and produce nitric oxide to facilitate accelerated wound healing [87]. Following VACV infection, Ly6G^+^ cells produce ROS, which can also act to increase wound healing [88], and ablation of ROS production following VACV infection causes a large increase in pathology [12]. Therefore, all of our previous observations, in which depletion or ablation of various components of the immune system has minor changes upon local virus replication, but substantial changes upon local tissue pathology, can be attributed to alterations in an early wound healing response.

Here, we initially investigated the role of γδ T cells in control of replication and spread of VACV, following cutaneous infection. γδ T cells have been implicated in the host response to VACV [7,8,9,10] and other poxviruses [56,57,58,59,60] after systemic infection. This protective effect has been attributed to cytolytic activity against VACV-infected cells [7,10], to γδ T cell-mediated production of IFN-γ [8,57], or to the ability of γδ T cells to modulate the T_CD8+_ response [11]. Here, following cutaneous infection with VACV, we show that epidermal Vγ3^+^ DETCs, which are a major population in mice but not in humans [89], acquire a GzB^+^ CD107a^+^ cytolytic phenotype. Neither the resident nor recruited dermal γδ T cell populations acquire this cytolytic phenotype. However, we, and others, [90] have found no difference in VACV replication in the skin of mice lacking γδ T cells, so any cytolytic contribution of these cells may be compensated for by infiltrating αβ T cells. None of the resident or recruited cutaneous γδ T cell populations appeared to be primed to make IFN-γ after VACV infection, even after in vitro restimulation, and there was a small increase in IFN-γ production in the skin of VACV-infected TCRδ^−/−^ mice compared to WT mice. However, KCs did make IFN-γ protein, shortly after VACV infection, and this contributed to control of swelling and, potentially subsequent pathology, but not to control of VACV replication. Therefore, the functions previously attributed to γδ T cells in control of virus replication and virus-induced pathology do not play a role in the pathology observed after cutaneous infection of TCRδ^−/−^ mice with VACV.

A sizeable portion of dermal γδ T cells produced IL-17A after cutaneous VACV infection, a cytokine that is required for efficient wound healing [25,91]. IL-17A mRNA was reduced to almost background levels in VACV-infected TCRδ^−/−^ mice 5d post-infection, a time point that precedes the later infiltration of T_CD4+_ that may produce IL-17A. Indeed, 8d post-infection, it was primarily T_CD8+_, not T_CD4+_, that produced IL-17A upon the restimulation of αβ T cells from infected skin. However, at early time points after infection, dermal γδ T cells may moderate wound healing, at least partially, via production of IL-17A in response to cutaneous VACV infection. γδ T cells have long been known to play a role in cutaneous wound healing after activation by KCs [31,32,33,34], where they migrate to a site of injury [27,35,36] and produce a number of cytokines that promote the wound healing response [37,38,39,40,41,42,43,44,45]. We did find a marked increase in VACV-induced pathology in the absence of γδ T cells, consistent with a role for these cells in establishment of the wound healing response, following VACV infection. However, IL-17 production following cutaneous poxvirus infection has also been implicated in inhibition of local NK cell activity, allowing the development of severe skin lesions in a mouse model of eczema vaccinatum [92]. Therefore, the role of IL-17, and particularly IL-17, production by γδ T cells following cutaneous VACV infection may be a nuanced balance between pro- and anti-wound healing responses that depends upon existing conditions in the skin.

Our data clearly demonstrate that γδ T cells modulated VACV-induced skin pathology. The enhanced pathology that resulted from VACV infection in the absence of γδ T cells prompted us to examine the range of wound healing-associated molecules induced by VACV infection that are discussed above. When we examined changes in the wound healing signature induced by cutaneous VACV infection in TCRδ^−/−^ mice, we originally anticipated that γδ T cells may alter the profile of chemokines that are induced. This is because γδ T cells have previously been reported to change recruitment of monocytes and neutrophils [27,28], including after VACV infection [90]. However, we found no discernable role for γδ T cells in modulation of chemokine or chemokine receptor expression after VACV infection and observed equivalent recruitment of myeloid cell populations in VACV-infected WT and TCRδ^−/−^ mice. The differences we observed from a previously published report, in which VACV-infected TCRδ^−/−^ mice exhibited less pathology than infected WT mice, likely because of alterations in neutrophil recruitment [90], may be attributable to the cutaneous microbiome in animal facilities. We have anecdotally observed a marked difference in both the magnitude and mechanisms involved in VACV-induced pathology in mice in the presence or absence of pathogenic bacteria, so we strive to ensure that mice are kept as pathogen-free as possible without rederivation into a germ-free facility. The composition of the skin microbiome has a profound effect upon wound healing [93] and future studies in germ-free animals reconstituted with different skin-resident bacteria will likely reveal roles for specific bacterial–host interactions after both “sterile” and virus-infected wounding.

When we examined the difference between the constitutive expression of wound healing-associated cytokines in WT and TCRδ^−/−^ mice, we found two genes that were downregulated in TCRδ^−/−^ mice, in the absence of virus infection. Expression of these two genes that encode IL-16 and adiponectin was also reduced in infected TCRδ^−/−^ mice vs. infected WT mice. IL-16 typically increases inflammation in the skin and inhibits the wound healing response [94], but adiponectin promotes wound healing by increasing KC proliferation and migration [95]. Therefore, deficits in γδ T cells may partially account for the previously described defect in sterile wound healing [33,45] via reduced expression of adiponectin.

Upon VACV infection of TCRδ^−/−^ mice, we found a similar response to that observed in WT mice, with upregulation of many of the mediators of wound healing that we had observed in WT mice and downregulation of a similar pattern of cytokines as well. However, there were two marked changes between VACV-infected TCRδ^−/−^ and WT mice, namely a failure to upregulate expression of the IL-10 family members, *il22* and *il10*, in TCRδ^−/−^ mice in response to VACV infection. IL-10 is produced after VACV infection by T_CD8+_ but is also induced prior to the infiltration of large numbers of these cells, indicating a role for γδ T cell-mediated production. However, although γδ T cells from the liver can produce IL-10 following bacterial infection [96], we found that γδ T cells do not produce IL-10 themselves, but induce production from other cells, likely skin-resident myeloid cells such as Langerhans cells and dermal DC (Figure 8M–O). IL-10 production drives skin regeneration, likely by altering the phenotype of macrophages, as the IL-10R is not expressed by KC [97]. Therefore, a reduction in IL-10 expression may drive part of the increase in pathology we observe following VACV infection of TCRδ^−/−^ mice. Such an increase in pathology may actually be detrimental to the virus, as some skin-tropic poxviruses encode an IL-10 homolog that enhances wound healing [98,99,100,101], indicating that IL-10 expression may be evolutionarily beneficial for the virus during natural skin infection, perhaps by inhibiting the chances of a competing local secondary bacterial infection.

Expression of IL-22 is often associated with inflammatory skin conditions, such as psoriasis [102], where it is often co-expressed with IL-17A [103]. Expression of both IL-22 and IL-17A is most often associated with Th17 T_CD4+_ cells, but both cytokines are produced in large quantities in cutaneous tissue by γδ T cells [104]. Although technical issues prevented us from establishing that the same populations of γδ T cells produce both IL-17A and IL-22 following cutaneous VACV infection, it is clear that the majority of these cytokines produced 5d post-VACV infection come from γδ T cells and these cytokines likely drive important components of the early wound healing response. IL-22 acts on dermal fibroblasts to drive expression of extracellular matrix proteins [105] and acts on KC to increase proliferation [102,106,107] via miR-197-driven mechanisms [108], as observed after bacterial skin infections [109]. In addition to the IL-10 family members, we also found that expression of both *fgf7* (encoding KC Growth factor) and *fgf9* were markedly reduced in VACV-infected TCRδ^−/−^ mice, consistent with reports of their production by γδ T cells and their roles in cutaneous wound healing [40,45]. Therefore, γδ T cells contribute in multiple ways to the cutaneous wound healing response following VACV infection.

In summary, we find here that neither resident nor recruited γδ T cells, nor cytokines produced by these cells, are involved in control of virus replication or spread following cutaneous infection. Rather, we describe here a uniquely configured wound healing response initiated in the skin of virus-infected mice, prior to the peak of virus replication and before adaptive immune mechanisms have been deployed successfully to clear the infection. We find that both resident and recruited γδ T cells are part of this induced wound healing response via production of IL-17A and IL-22, as well as the induction of IL-10 in other cells, and that a deficit in γδ T cells causes a profound increase in tissue pathology following infection. These findings are important in understanding how wound healing is mediated following a cutaneous virus infection in comparison to the paradigm of sterile wound healing. A prompt and appropriately controlled wound healing response is crucial to prevent secondary bacterial infections that could be deleterious for the virus-infected host, but also potentially for the virus itself.

## Figures and Tables

**Figure 1 viruses-16-00425-f001:**
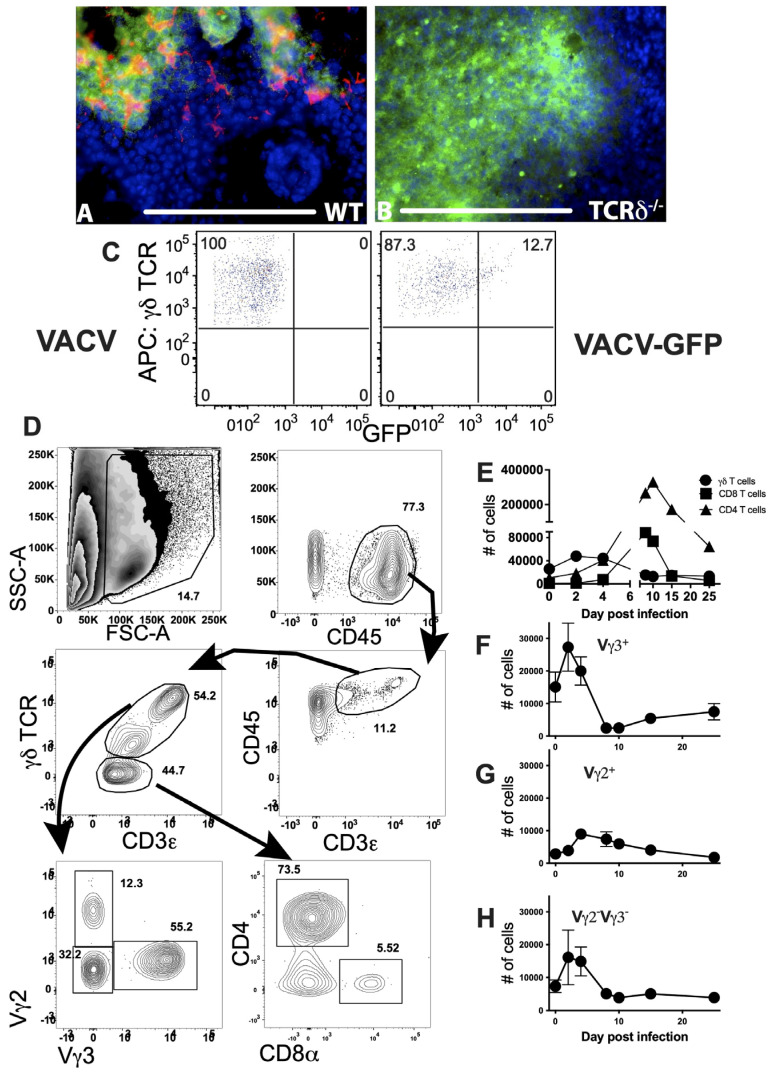
**Infected and uninfected γδ T cells localize within VACV-infected foci in the skin.** WT (**A**) and TCRδ^−/−^ (**B**) mice were infected with 10^4^ PFU of VACV-GFP using a bifurcated needle. Four days p.i., ears were harvested, frozen, and sections were stained with antibody to TCRδ (red). Infected GFP+ cells are visualized in green. Scale bar = 100 μm. (**C**) To confirm that VACV infected γδ T cells, we infected mice i.d. with 10^4^ PFU of VACV or VACV-GFP then harvested and digested ears for flow cytometric analysis 4d p.i. A proportion of γδ T cells, identified as CD45^+^, CD3^+^, and γδ TCR^+^, showed clear GFP fluorescence. (**D**) For all subsequent analyses, debris and dead cells were first excluded using a “live” cell gate. Singlets were identified by side scatter area (SSC-A) vs. forward scatter width (FSC-W) and then lymphocytes gated according to light scatter area. Within CD45^+^ cells, CD3ε^+^ T cells were subdivided into γδ T cells (TCRδ^+^) and αβ T cells (TCR δ^−^), which include T_CD4+_ and T_CD8+_. γδ T cell subsets in the skin were identified as follows: Vγ3^+^ dendritic epidermal T cells (DETCs) and Vγ2^+^ or Vγ2^−^Vγ3^−^ dermal γδ T cells. Data depict T cells in ears of WT mice 4d p.i. with VACV and are representative. (**E**,**F**) WT mice were infected i.d. with 10^4^ PFU VACV. Cells were harvested from whole ears of mice on day 0 (uninfected), 2, 4, 8, 10, 15, and 25 p.i. and T cell responses were monitored using flow cytometry. (**E**) Numbers of total γδ T cells (CD3ε^+^TCRδ^+^, black circles), T_CD4+_ (CD3ε^+^TCRδ^−^CD4^+^, gray squares), and T_CD8+_ (CD3ε^+^TCRδ^−^CD8α^+^, triangles) cells per pair of uninfected and infected ears. Numbers of epidermal-resident Vγ3^+^ DETCs (**F**), Vγ2^+^, (**G**) Vγ2^−^Vγ3^−^, or (**H**) dermal γδ T cells (black squares and diamonds, respectively) in ears after i.d. VACV infection. Values at each time point represent the mean ± SEM in 6–10 pairs of ears from three independent experiments.

**Figure 2 viruses-16-00425-f002:**
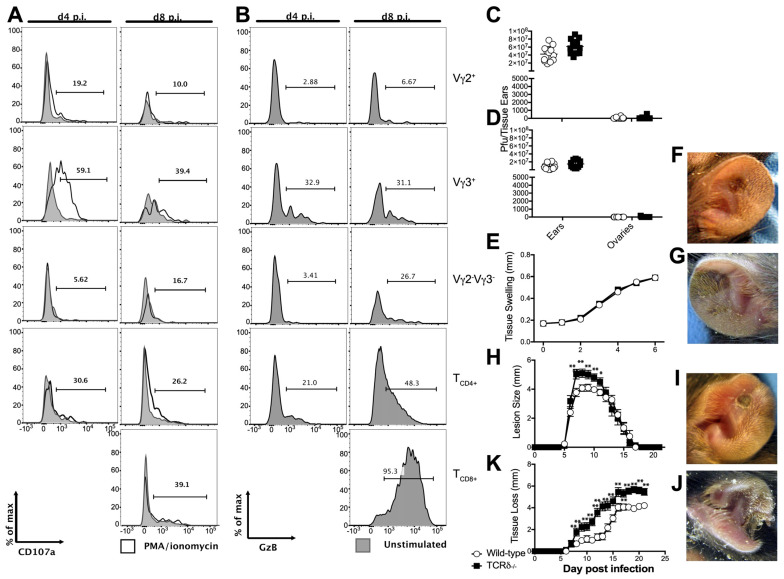
**Infected TCRδ^−/−^ mice display enhanced tissue pathology**, **but no difference in local VACV replication or systemic virus spread.** WT (**A**–**F**,**H**,**I**,**K**) or TCRδ^−/−^ (**C**–**E**,**G**,**H**,**J**,**K**) mice were infected i.d. in the ear pinnae with 10^4^ PFU of VACV and ears were harvested and dissociated for analysis on day four (**A**,**B**) or day eight (as shown, (**A**,**B**)) post-infection. Gating strategies are as shown in Figure 1. Potential cytolytic function was measured using the cell surface expression of CD107a (LAMP1) in response to activation with PMA-ionomycin (**A**), or intracellular expression of granzyme B (**B**) by Vγ3^+^ DETCs, Vγ2^+^ or Vγ2^−^Vγ3^−^ dermal γδ T cells, T_CD4+,_ or T_CD8+_ 4d or 8d post-infection. T_CD8+_ were undetectable at the site of infection 4d post-infection, so are only shown at 8d post-infection. (**C**–**E**,**H**,**K**) WT (circles) or TCRδ ^−/−^ (squares) mice were infected i.d. in the ear pinnae with 10^4^ PFU of VACV. On day five (**C**) and day eight (**D**) p.i., pairs of ears and ovaries were harvested from infected WT and TCRδ^−/−^ mice to measure titers of VACV by plaque assay. Data are representative of seven pairs of ovaries and ears per group from two independent experiments. Tissue swelling (**E**) was assessed 6d after infection. The appearance of lesions in WT (**F**) or TCRδ^−/−^ (**G**) mice was visualized 8d post-infection and lesion size was quantified (**H**) for 21 days post infection. The ensuing tissue loss in WT (**I**) or TCRδ^−/−^ (**J**) mice was visualized 14d post-infection and quantified (**K**) for 21 days post-infection. Data in (**E**,**H**,**K**) depict the mean ± SEM of 30 ears per group from three independent experiments. * *p* < 0.05, ** *p* < 0.005 using Student’s *t*-test.

**Figure 3 viruses-16-00425-f003:**
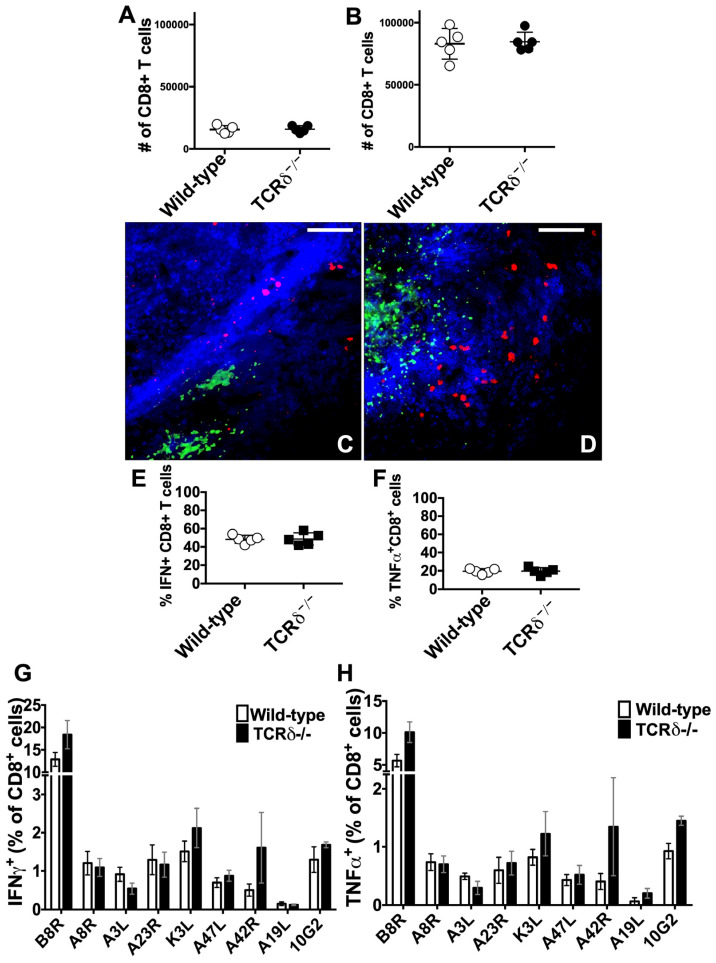
**T_CD8+_ recruitment, localization, and function are not affected in the absence of γδ T cells.** WT or TCRδ^−/−^ mice were infected i.d. with 10^4^ PFU VACV per ear and cells harvested to analyze the recruitment, localization, and function of T_CD8+_. Ears were harvested 5d (**A**) or 8d post-infection (**B**–**H**) and the number of T_CD8+_ (**A**,**B**), localization of T_CD8+_ (in red) relative to VACV infected cells (in green) (**C**,**D**), number of T_CD8+_ in the VACV-infected ear producing IFN-γ (**E**) or TNFα (**F**) directly ex vivo, or the proportion of splenic T_CD8+_ producing IFN-γ (**G**) or TNFα (**H**) in response to stimulation with VACV-derived MHC Class I binding epitopes was investigated. Scale bar = 100 μm.

**Figure 4 viruses-16-00425-f004:**
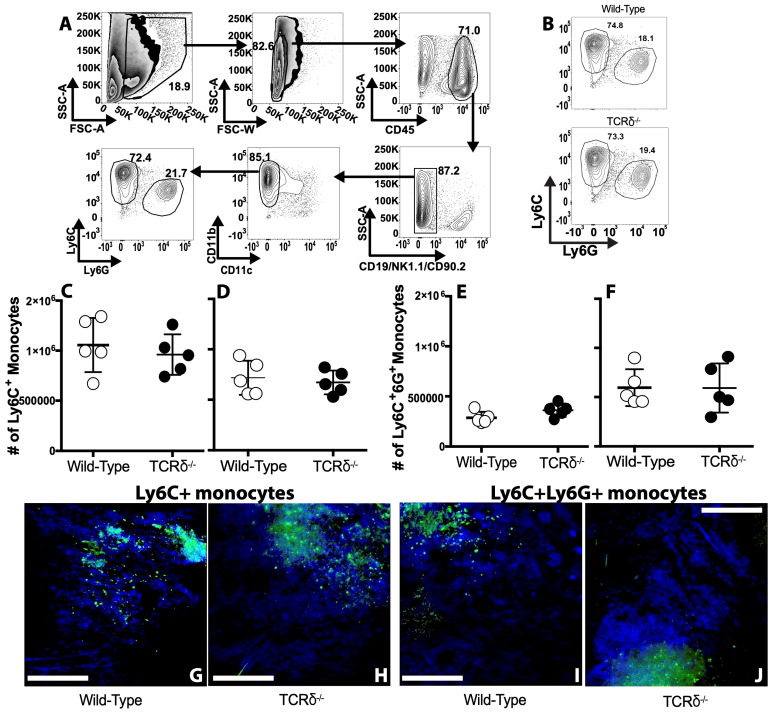
**Monocyte recruitment and localization is normal in mice lacking γδ T cells.** WT and TCRδ^−/−^ mice were infected i.d. in each ear pinnae with VACV. The gating strategy is shown in (**A**). As in Figure 1, we first excluded debris and dead cells by scatter area and then gated singlets by scatter area vs. width. B cells, T cells, non-NK innate lymphoid cells (ILCs), and NK cells were “dumped” from CD45^+^ cells by excluding CD19^+^, CD90.2^+^, and NK1.1^+^ cells. CD45^+^CD19^−^CD90.2^−^NK1.1^−^CD11c^−^CD11b^+^ monocyte/macrophages were subdivided into Ly6C^+^Ly6G^−^ classical inflammatory monocytes and Ly6C^Int^Ly6G^+^ regulatory myeloid cells. Representative data from a WT mouse 5d p.i. are shown in (**B**). (**C**–**F**) Cells were harvested from VACV-infected WT or TCRδ^−/−^ mice to analyze the recruitment of Ly6C^+^Ly6G^−^ and Ly6C^+^Ly6G^+^ monocyte populations. (**C**,**D**) Quantification of Ly6C^+^Ly6G^−^ inflammatory monocytes 5d (**C**) and 8d (**D**) post-infection in WT vs. TCRδ^−/−^ mice. (**E**,**F**) Quantification of Ly6C^+^Ly6G^+^ monocytes 5d (**E**) and 8d (**F**) post-infection in WT vs. TCRδ^−/−^ mice. All values depict the number of cells per pair of ears. Data in (**B**–**F**) are representative of six pairs of ears per group from three independent experiments. (**G**–**J**) WT (**G**,**I**) or TCRδ^−/−^ (**H**,**J**) mice were infected i.d. with 10^4^ PFU VACV-GFP per ear and ears were harvested and frozen 8d post-infection. Tissue sections were cut, stained with antibodies to GFP (in green) and either Ly6C (**G**,**H**) or Ly6G (**I**,**J**) (in blue), and were visualized using deconvolution microscopy. Each image is representative of four ears per group from two experiments. Scale bar = 100 μm.

**Figure 5 viruses-16-00425-f005:**
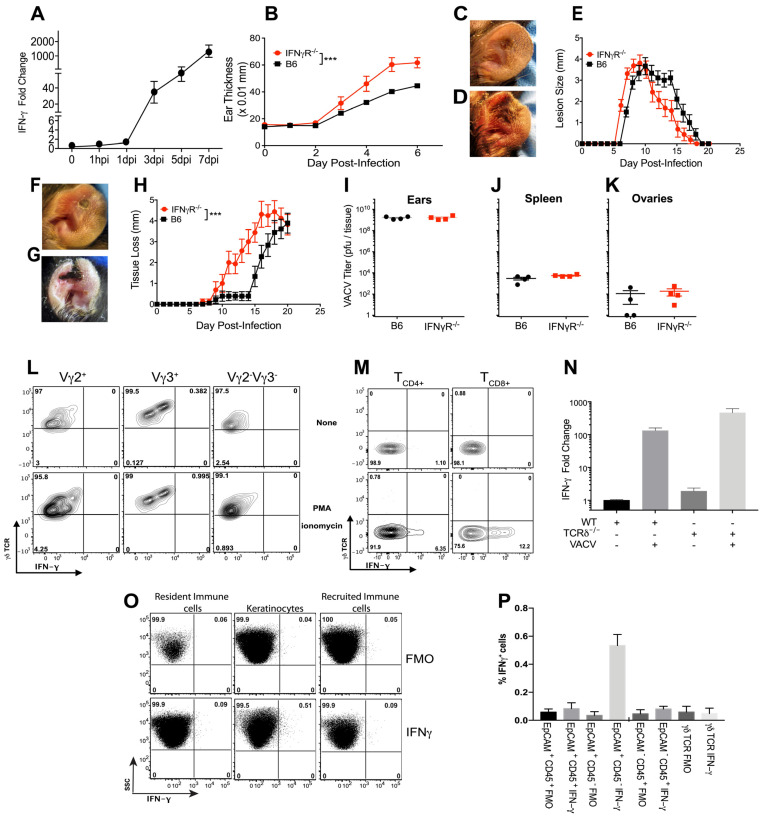
**IFN*-γ* produced early after VACV infection, independent of γδ T cells, is required to control local tissue pathology but not local VACV replication or systemic virus spread.** WT (**A**–**C**,**E**,**F**,**H**–**P**), TCRδ^−/−^ (**N**), or IFN-γR^−/−^ (**B**,**D**,**E**,**G**,**H**–**K**) mice were infected i.d. in the ear pinnae with 10^4^ PFU of VACV and ears harvested and dissociated for analysis. (**A**) Time course of IFN-γ mRNA expression, measured using qPCR, in the ear after VACV infection. Data are representative of three independent experiments, showing the mean ± SEM of three or four biological replicates in each. Tissue swelling (**B**) was assessed for 6d post-infection. The appearance of lesions in WT (**C**) or IFN-γR^−/−^ (**D**) mice was visualized 8d post-infection and lesion size was quantified (**E**) for 21d post-infection. The ensuing tissue loss in WT (**F**) or IFN-γR^−/−^ (**G**) mice was visualized 14d after infection and quantified (**H**) for 21d post-infection. Data in (**B**,**E**,**H**) depict the mean ± SEM of 30 ears per group from three independent experiments. Five days (**I**–**K**) post-infection, pairs of ears (**I**), spleen (**J**), and ovaries (**K**) were harvested from infected WT and IFN-γR^−/−^ mice to measure titers of VACV by plaque assay. Data are representative of seven pairs of ovaries and ears and seven spleens per group from three independent experiments. (**L**) The production of IFN-γ by Vγ3^+^ DETCs, Vγ2^+^ cells, or Vγ2^−^Vγ3^−^ dermal γδ T cells from ear (gated as in Figure 1) or by lymph node T_CD4+_ or T_CD8+_ (**M**) 4d post-VACV infection in the absence of PMA-ionomycin activation. Data in L and M are representative of twelve separate biological replicates from three independent experiments. (**N**) The relative IFN-γ mRNA expression in the ear of WT or TCRδ^−/−^ mice 4d after i.d. VACV infection. Data show the mean ± SEM from three independent experiments, with three biological replicates in each. Representative data (**O**) and compiled and quantified total data from three independent experiments, with three biological replicates in each (**P**) of intracellular cytokine staining of IFN-γ within EpCAM^+^CD45^−^ KC, EpCAM^+^CD45^+^ resident immune cells, and EpCAM^−^CD45^+^ recruited immune populations. *** *p* < 0.0001 using Student’s *t*-test.

**Figure 6 viruses-16-00425-f006:**
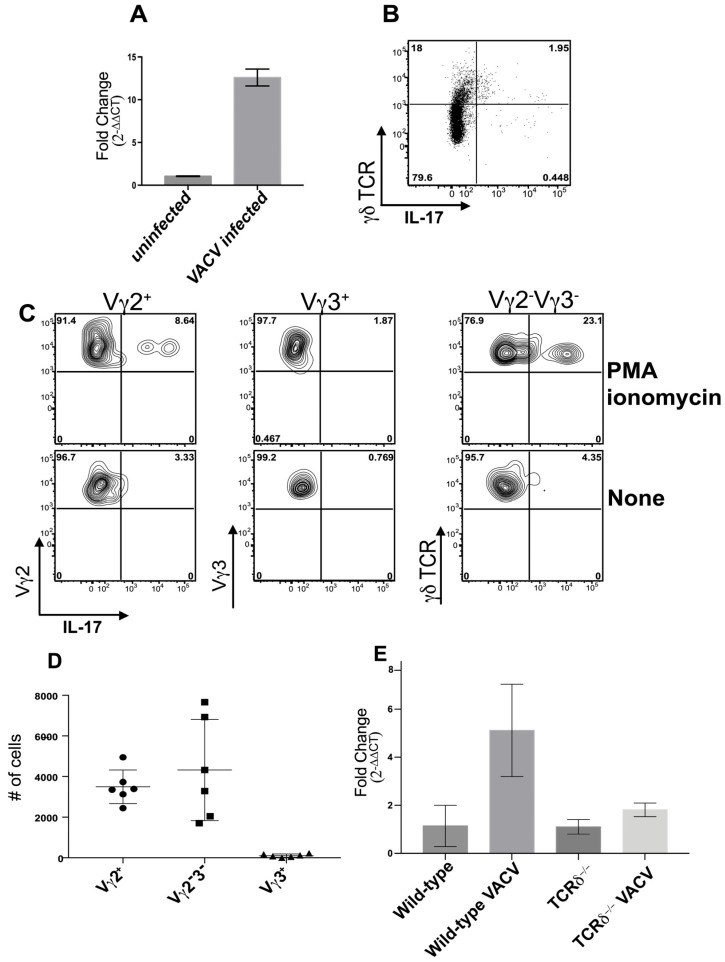
**Specific subsets of γδ T cells produce IL-17A upon cutaneous VACV infection.** WT (**A**–**E**) or TCRδ^−/−^ (**E**) mice were infected i.d. in the ear pinnae with 10^4^ PFU of VACV. (**A**) Expression of *il17a* mRNA in the ear of VACV-infected or uninfected WT mice. (**B**–**D**) Gating strategies are as shown in Figure 1. Production of IL-17A by γδ T cells (**B**), or by Vγ3^+^ DETCs, Vγ2^+^ or Vγ2^−^Vγ3^−^ dermal γδ T cells (**C**) 4d post-VACV infection in the absence of PMA-ionomycin activation. (**D**) Quantitation of the number of cells displaying intracellular cytokine staining of IL-17A within Vγ3^+^ DETCs, Vγ2^+^ or Vγ2^−^Vγ3^−^ dermal γδ T cells populations 4d post-infection. (**E**) Expression of *il17a* mRNA in the ear of VACV-infected or uninfected WT or TCRδ^−/−^ mice, as shown, 5d post-infection. Data are representative of twelve separate biological replicates from three independent experiments (**A**–**E**).

**Figure 7 viruses-16-00425-f007:**
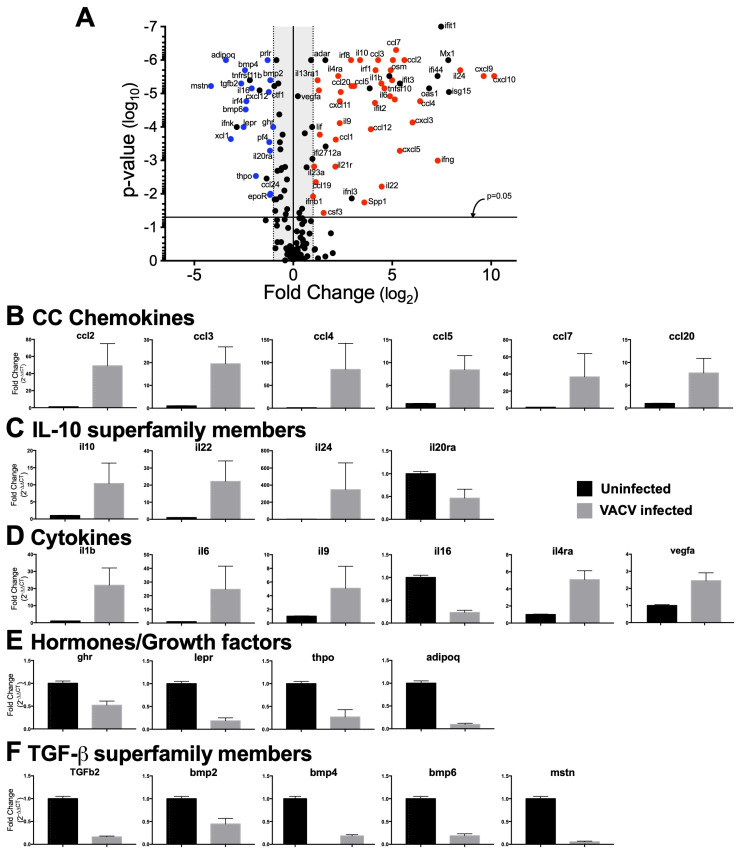
**A wound healing signature is induced in the ear of VACV-infected mice**. WT mice were infected i.d. in the ear pinnae with 10^4^ PFU of VACV, ears were harvested 5d post-infection and mRNA levels of target molecules measured using Qiagen qPCR array plates, as described in the methods sections. (**A**) Expression of cytokines, chemokines, interferons, and receptors displayed using volcano plots, which demonstrate statistically significant points if above the *p* = 0.05 line. Gene expression changes in WT mice in response to VACV, with genes that are upregulated >2-fold by VACV infection in the upper right-hand quadrant and genes that are downregulated > 2-fold by VACV infection in the upper left-hand quadrant. Genes that are significantly (>2-fold, *p* < 0.05) upregulated and have a defined role in wound healing are shown in red, and those that are significantly (>2-fold, *p* < 0.05) downregulated and have a defined role in wound healing are shown in blue. (**B**–**F**) Representative plots from the data displayed in (**A**), showing changes in mRNA levels 5d post-VACV infection in genes associated with wound healing, including CC chemokines (**B**), IL-10 superfamily members (**C**), other cytokines, and cytokine receptor antagonists (**D**), growth factors, hormones, and their receptors (**E**) and TGF-β superfamily members (**F**). n = 28 naïve, 29 VACV-infected mice per group.

**Figure 8 viruses-16-00425-f008:**
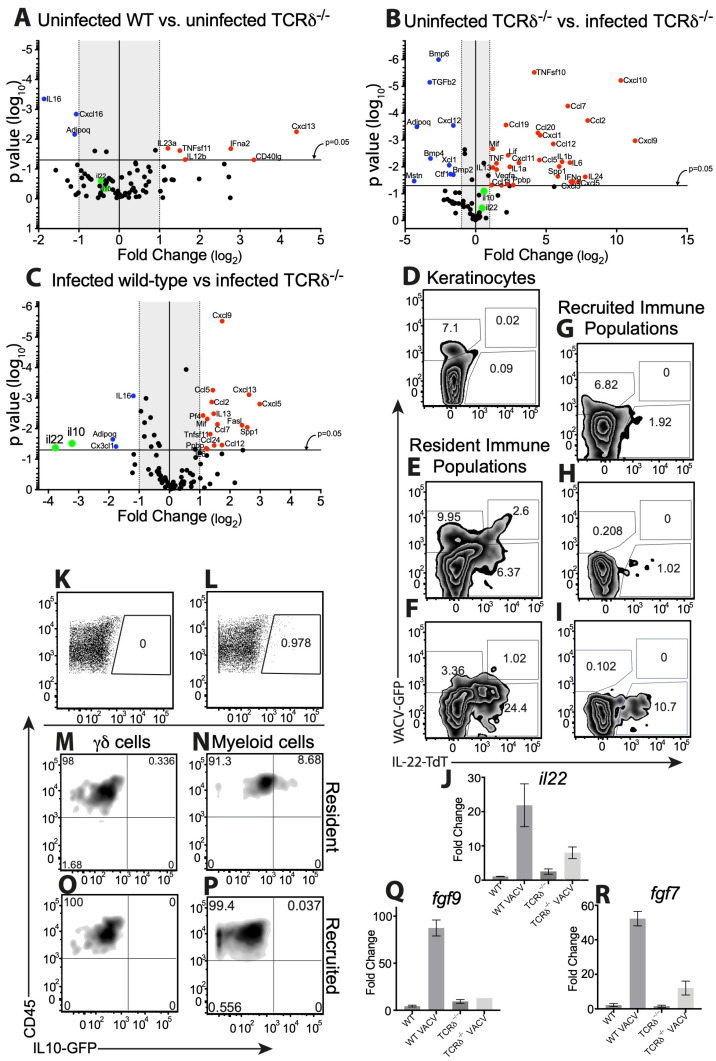
**Expression of wound healing molecules is altered in the absence of γδ T cells.** WT or TCRδ^−/−^ (**A**–**C**,**J**,**P**,**R**), IL-22-TdTomato reporter (**D**–**I**), or IL-10GFP reporter (**K**–**P**) mice were infected with VACV, ears were harvested 5d post-infection, and (**A**–**C**) mRNA levels of target molecules, production of IL-22 (**D**–**I**) or IL-10 (**K**–**P**) by resident or recruited cell populations from the ear, or bulk mRNA expression levels of il22 (**J**), fgf9 (**Q**), or fgf7 (KGF, **R**) were measured. (**A**–**C**) Volcano plots of cytokine and chemokine expression, with statistically significant points above the *p* = 0.05 line. (**A**) Cytokines and chemokines regulated in uninfected TCRδ^−/−^ mice relative to WT mice, showing genes that are upregulated in uninfected TCRδ^−/−^ mice in the upper right-hand quadrant and downregulated genes in uninfected TCRδ^−/−^ mice in the upper left-hand quadrant. (**B**) Gene expression changes in VACV-infected TCRδ^−/−^ mice relative to uninfected TCRδ^−/−^ mice. (**C**) Gene expression changes in VACV-infected TCRδ^−/−^ mice relative to VACV-infected WT mice. (**A**–**C**) Genes that are downregulated in VACV-infected TCRδ^−/−^ mice relative to VACV-infected WT mice (*il10* and *il22*) are displayed in green to facilitate ease of recognition in each plot. (**D**–**I**) Flow cytometry of IL-22-TdTomato reporter mouse ears infected with VACV-GFP, stained to identify resident or recruited cell populations. Density plots show infection (marked by VACV-GFP) vs. IL-22 production (marked by TdTomato expression) within EpCAM^+^CD45^−^ KC (**D**), resident immune cells (EpCAM^+^CD45^+^, (**E**)), resident γδ T cells (EpCAM^+^CD45^+^γδTCR^+^, (**F**)), recruited immune cell populations (EpCAM^−^CD45^+^, (**G**)), recruited lymphocytes (EpCAM^−^CD45^+^CD11b^−^γδ TCR^−^, (**H**)), and recruited γδ T cells (EpCAM^−^CD45^+^γδTCR^+^, (**I**)). (**J**) Expression of *il22* in ear tissue of uninfected or VACV-infected WT or TCRδ^−/−^ mice 5d p.i. measured using RT-qPCR. (**K**–**P**) Ears of WT (**K**) or IL-10-GFP reporter (**L**–**P**) mice infected with VACV then stained to identify resident or recruited cell populations. Dot plots show IL-10 expression (marked by GFP) in CD45+ from WT (**K**) or IL-10-GFP reporter mice (**L**). Density plots show CD45 vs. IL-10 production (marked by GFP) within resident (EpCAM^+^CD45^+^, (**M**,**N**)) or recruited (EpCAM^−^CD45^+^, (**O**,**P**)) immune populations, separated into γδ T cells (γδTCR^+^, (**M**,**O**)) and myeloid cell populations (CD11b, γδTCR^−^, **N**,**P**). (**Q**,**R**) Expression of *fgf9* (**Q**) or *fgf7*(**R**) (which encodes KGF) in ear tissue of uninfected or VACV-infected WT or TCRδ^−/−^ mice 5d p.i. measured using RT-qPCR.

## Data Availability

All data associated with this work will be deposited under license in the Penn State open access repository Scholarsphere at https://scholarsphere.psu.edu/ (accessed on 5 February 2024).

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
