# Peer review of "γδ T Cells Mediate a Requisite Portion of a Wound Healing Response Triggered by Cutaneous Poxvirus Infection"

_viruses, 2024, doi:10.3390/v16030425_

Round 1

Reviewer 1 Report

Comments and Suggestions for Authors

In the manuscript “viruses-2885978” the authors report on the roles of murine gamma/delta T cells in vaccinia virus infection and wound healing. A lot of data is presented that shows that gamma/delta T cells influenced wound healing during a local infection but did not affect virus replication or dissemination. Experiments were carried out well and the appropriate controls were included. The interpretations of the presented data are appropriate. Overall, the study adds novel and important information and is of high interest to both the poxvirus and immunology fields.

My only issue is that there are a few instances where figures were mislabeled/mixed up or some information was missing (see below). Because the presented data is very complex, the authors should carefully check the entire manuscript, in case there are more similar issues that escaped the reviewers.

Fig. 2: It is hard to differentiate the symbols for Fig. 2 C-K. The use of color for wild type and TCRd-/- would make it easier to distinguish between the two. Statistical differences for lesions and tissue loss is mentioned in results, and significance levels are described in the figure legend, but no significance values/asterisks are shown Fig. 2H,K.

Fig. 3: For consistency, it would be better if the same symbols for TCRd-/- (squares) were used here, as in Fig. 2. Same goes for figure 4.

Fig. 7: The colors representing uninfected (grey) and VACV infected (black) are mixed up. Infected should be grey. This needs to be corrected. The authors write in the results section: ” Growth factors such as vascular endothelial growth factor (vegf), Leukemia inhibitory factor (lif), and osteopontin (spp1) were also upregulated by VACV infection (Fig. 7E).” This is not shown in Fig. 7E.

Author Response

We have changed the symbols in Figures 2, 3 and 4 and added symbols to indicate significance values in Figure 2.  We apologize for the mislabeling in Figure 7 and have corrected this.

Reviewer 2 Report

Comments and Suggestions for Authors

This paper by Reider, et al. studied the possible role of gamma/delta T cells following a cutaneous poxvirus infection. I found the study to be novel and interesting. The authors should note that the paper is on the long side and could easily be condensed by removing superfluous text.

Please consider the suggestions below to improve the manuscript:

Methods (line 94): I could find no description of how lesion progression and tissue loss were actually measured/quantified. Please add in a description of this process.

Results (line 196): Vgamma 2 and 3 were stained directly but not Vgamma 4. I assume there is no commercially available antibody to stain Vgamma4. If so, please state this directly so the reader knows why the paper continuously refers to Vgamma2 minus, Vgamma3 minus cells.

Results (line 197): The authors mention about expanded vs recruited populations of gamma/delta T cells in Figure 1. However, I see no formal analysis in the figure to differentiate between the two. For example, I do not see Ep-CAM measured in Figure 1 or mentioned in the corresponding text.

Results, Figure 1A and 1B: Please indicate the magnification and/or provide a scale bar on the images. This comment applies to the other IF images as well. Additionally, there appears to be much more green signal (VACV) in panel B compared to panel A but data in Figures 2C and 2D show only small differences in viral load when quantified with a plaque assay. Please comment and/or rectify.

Results, Figure 1F-H: I suggest adding more tick marks on the x-axis to help better delineate the days post-infection. Additionally, I noticed that there is no sham or mock control here. Can the authors comment on what would be expected with respect to gamma/delta T cell levels if 10 uL of saline alone (no virus) were injected into the ear pinna? It seems to me a mock control should be shown here at least once but is it necessary? Why or why not?

Figure 2: The legend references statistical analysis being performed but no asterisks appear in the figure panels. Please add those where needed and also indicate no statistical difference where appropriate as well.

Figures 3 and 4 in my version are out of order and the legends don't match.

current Figure 4E and 4F (the CD8 T cell data): IFNg is shown at day 5 and TNFa for day 8. Why not just show both cytokines at both time points?

Figure 7: I think the bar colors are labeled backwards. It seems from the text (e.g., lines 418-421) that uninfected should be black and VACV-infected grey. Please check.

Figure 8A-C: Add y-xis labels.

Final comment and critique: This paper is certainly interesting. However, in my view, it was mostly observational in nature but less robust in describing a mechanism. For example, the authors implicate several molecules that may explain the observed results in wound healing and tissue damage in the absence of gamma/delta T cells. It would be nice to know which of these, if any, is the "major player" to explain the phenotype. Or perhaps it's not just one cytokine that is most important.

Author Response

Changes to the text are highlighted.

A description of the simple methodology used to measure lesion size and tissue loss has been added (now line 95).

There is no commercially available antibody for mouse V gamma 4.  text to that affect has no been added (like 200).

The text concerning expansion or recruitment has been modified (line 201), as there is no way to ascertain a source of cells without performing bone marrow chimera experiments.

Images in Figure 1, 3 and 4 now have scale bars.

Text has been added on line 176 to describe that the extent of the GFP+ lesion does not quantify infection in the ear, only the extent of the lesion in the image.

Text concerning a lack of increase in gamma delta T cell numbers after injection of diluent alone has been added on Line 186.

In Figure 2 indications of statistical significance have now been added.

Figures 3 and 4 had been correctly positioned in a previous submission.

In Figure 3 E-H all of the data are from day 8 of infection, not d5 vs d8 as indicated by the reviewer.

Figure 7 - we apologize for our error and the bars are now correctly labeled.

Figure 8 - y-axis labels have now been added.

We agree that we have not observed a single function of gamma delta T cells that accounts for the phenotype observed.  However, gamma delta T cells are multifunctional and it is likely that more than one mechanism, including direct and indirect mechanisms, accounts for the increased damage we observe.